# Combination of Roll Grinding and High-Pressure Homogenization Can Prepare Stable Bicelles for Drug Delivery

**DOI:** 10.3390/nano8120998

**Published:** 2018-12-03

**Authors:** Seira Matsuo, Kenjirou Higashi, Kunikazu Moribe, Shin-ichiro Kimura, Shigeru Itai, Hiromu Kondo, Yasunori Iwao

**Affiliations:** 1Department of Pharmaceutical Engineering, School of Pharmaceutical Sciences, University of Shizuoka, 52-1 Yada, Suruga-ku, Shizuoka 422-8526, Japan; s17119@u-shizuoka-ken.ac.jp (S.M.); s-kimura@u-shizuoka-ken.ac.jp (S.-i.K.); s-itai@u-shizuoka-ken.ac.jp (S.I.); hkondo@u-shizuoka-ken.ac.jp (H.K.); 2Graduate School of Pharmaceutical Sciences, Chiba University, 1-8-1, Inohana, Chuo-ku, Chiba 260-8675, Japan; ken-h@faculty.chiba-u.jp (K.H.); moribe@faculty.chiba-u.jp (K.M.)

**Keywords:** lipid nanoparticles, cryo transmission electron microscopy, atomic force microscopy, bicelle, micelle, liposome

## Abstract

To improve the solubility of the drug nifedipine (NI), NI-encapsulated lipid-based nanoparticles (NI-LNs) have been prepared from neutral hydrogenated soybean phosphatidylcholine and negatively charged dipalmitoylphosphatidylglycerol at a molar ratio of 5/1 using by roll grinding and high-pressure homogenization. The NI-LNs exhibited high entrapment efficiency, long-term stability, and enhanced NI bioavailability. To better understand their structures, cryo transmission electron microscopy and atomic force microscopy were performed in the present study. Imaging from both instruments revealed that the NI-LNs were bicelles. Structures prepared with a different drug (phenytoin) or with phospholipids (dimyristoylphosphatidylcholine, dipalmitoylphosphatidylcholine, and distearoylphosphatidylcholine) were also bicelles. Long-term storage, freeze-drying, and high-pressure homogenization did not affect the structures; however, different lipid ratios, or the presence of cholesterol, did result in liposomes (5/0) or micelles (0/5) with different physicochemical properties and stabilities. Considering the result of long-term stability, standard NI-LN bicelles (5/1) showed the most long-term stabilities, providing a useful preparation method for stable bicelles for drug delivery.

## 1. Introduction

About 40% of marketed drugs and 70% or more of drug candidates have exhibited poor water solubility [1]. These drugs also have problems such as non-constant absorption in vivo and low bioavailability (BA) [2]. Approaches to improve solubility and absorption include drug miniaturization [3,4,5], amorphization [6,7,8], and solid dispersion [9,10,11]. In particular, nanoparticle formulations have been useful in drug delivery [12]. As the particle size decreases, the dissolution rate increases because of the increased surface area, according to the Noyes–Whitney equation [13,14]; and solubility increases according to the Ostwald–Freundlich equation [15]. In addition, nanoparticle formulations can pass through mucosal layers that can be a barrier to drug absorption, resulting in enhanced oral BA [16,17]. Ideal drug carriers are composed of biocompatible and biodegradable materials with a small size, a high drug-loading capacity, high stability, and easily modified surfaces. They include emulsions [18], liposomes [19], micelles [20], lipid nanoparticles [21], polymer nanoparticles [22], hydrogels [23], and carbon nanotubes [24]. Lipid nanoparticles for drug delivery systems (DDS) have been used for various dosage preparations such as oral, parenteral, dermal, pulmonary, rectal, and ocular [25]. 

Previously, we reported that the water solubility of nifedipine (NI, class II in the biopharmaceutics classification system [26]) was increased by encapsulation in lipid-based nanoparticle (NI-LN) suspensions. These were fabricated with a neutral phospholipid (hydrogenated soybean phosphatidylcholine, HSPC), and a negatively charged phospholipid (dipalmitoylphosphatidylglycerol, DPPG), using co-grinding by roll milling and subsequent high-pressure homogenization [27]. The mean particle size was 50 nm, with a narrow particle distribution (polydispersity index: PDI) of less than 0.3. In addition, they were stable for four months in cool, dark storage [28]. To test long-term stability, freeze-drying with sugar was performed; the mean particle size was maintained after reconstitution [28]. Furthermore, when NI-LN suspensions were administrated orally to rats, the area under the curve increased fourfold relative to that for NI suspensions, and the oral BA of NI (59%) was higher than that of NI bulk (50%) [29]. This LN preparation could increase the solubility of other water-insoluble drugs as well, regardless of the drug structure. Compared with other reported lipid nanoparticles, they had superior long-term stability without water-dispersible stabilizers, and high entrapment efficiency (EE) for many drugs [30,31,32,33,34]. 

Here, cryo transmission electron microscopy (cryo-TEM) and atomic force microscopy (AFM) imaging revealed that the NI-LNs were found to be bicelles, which are flat, disk-like lipid bilayers. Generally, bicelles are usually prepared from two phospholipids having different alkyl chain lengths (bilayers from long-chain lipids and edges from short-chain lipids) that are dissolved in organic solvent, evaporated, and hydrated [35,36]. Here, the NI-LNs used HSPC and DPPG that have similar acyl chain lengths, and were prepared without organic solvents via a combination of co-grinding with a roll mill and high-pressure homogenization. According to the structural analysis of lipid nanoparticles prepared under various conditions, the lipid ratios strongly affected the structures. The simple preparation for stable bicelles will be potentially useful for future DDS.

## 2. Materials and Methods 

### 2.1. Materials 

HSPC (COATSOME^®^ NC-21), and 1,2-distearoyl-sn-glycero-3-phosphocholine (DSPC, COATSOME^®^ MC-8080) were provided by Nippon Fine Chemical Co., Ltd. (Osaka, Japan). DPPG, sodium salt (COATSOME^®^ MG-6060LS), 1,2-dimyristoyl-sn-glycero-3-phosphocholine (DMPC, COATSOME^®^ MC-4040), and 1,2-dipalmitoyl-sn-glycero-3-phosphocholine (DPPC, COATSOME^®^ MC-6060) were purchased from Nippon Oil and Fats Co., Ltd. (Tokyo, Japan). NI (Japanese Pharmacopeia XVII) and cholesterol (Chol) were purchased by Sigma-Aldrich, Co. (St. Louis, MO, USA). Methanol (HPLC grade), formic acid (HPLC grade), and sucrose were purchased from Wako Pure Chemical Industries, Ltd (Osaka, Japan). The membrane filters (pore size: 0.20-μm) were purchased from Toyo Roshi Kaisha Ltd. (Tokyo, Japan). Phenytoin (PHT, Japanese Pharmacopeia XVII) was purchased from Sumitomo Dainippon Pharma Co., Ltd. (Osaka, Japan). All of the reagents were of the highest grade commercially available, and all of the solutions were prepared using deionized distilled water.

### 2.2. Preparation of Drug–Lipid Nanoparticle Suspensions

The drug (NI or PHT)-lipid nanoparticle suspensions were prepared as described previously with a slight modification [27]. Briefly, 40 mg of NI or PHT and 1000 mg of lipid (HSPC/DPPG (5/1) for standard NI-LNs, or (5/0), (5/0.5), and (0/5) molar ratios), were added to a mortar and physically mixed with a pestle. The mixture was then co-ground for five minutes with a roll mill (Model: R3-1R, Kodaira Seisakusho Co., Ltd., Tokyo, Japan) having three grinding rollers rotating at velocity ratios of 1/2.5/5.8. The sample mostly adhered to the rollers, but the mill was stopped every 30 seconds to collect fallen samples. The co-grinding cycle was repeated 10 times, and the mixture was then dispersed in 200 mL of distilled water and premixed with a Speed Stabilizer^®^ (Kinematica Co., Luzern, Switzerland) at 9000 rpm for 10 min. The suspension was then subjected to high-pressure homogenization (Microfluidizer^®^, M110-E/H; Microfluidics, Co., Newton, MA, USA) with a pass cycle of 100 at 175 MPa. The NI-LN suspension was filtered by suction aspirator with a 0.20-µm polytetrafluoroethylene or cellulose acetate membrane filter (Toyo Roshi Kaisha Ltd., Tokyo, Japan) to remove the particles whose size was over 200 nm. LNs with no drug loading were prepared in the same way. 

### 2.3. Freeze-Drying and Reconstitution of NI–Lipid Nanoparticles

Freeze-drying was performed with two mL of the NI-LN suspensions placed in a vial containing 100 mg of sucrose (5%, *w/v*) as a cryoprotectant. Sucrose was selected based on previous reports [28,37]. The vial was frozen at −70 °C overnight and then freeze-dried in a glass chamber for 48 h with a vacuum pump equipped with a vapor condenser (−20 °C, 0.0225 Torr). Reconstitution was performed immediately after the freeze-drying; two mL of deionized distilled water was added to the vial, and the sample was rehydrated via vortex agitation.

### 2.4. Measurement of Mean Particle Size and Zeta Potential

The mean particle size and the zeta potential of LN suspensions were measured with dynamic light scattering (DLS) (Zetasizer nano ZS, Malvern Instruments Ltd., Worcestershire, UK) at a scattering angle of 90° at room temperature (25 °C). Mean particle sizes were based on the scattering intensity, while the zeta potential was based on electrophoretic mobility. Since the values of mean count rate of the samples were 200–500 kcps, which are suitable for this measurement, intact samples without dilution were used in this measurement. We repeated measurement three times per sample. 

### 2.5. Measurement of the Concentration and the Entrapment Efficiency of Drugs in Drug–Lipid Nanoparticle Suspensions

The filtered lipid nanoparticles were defined as the soluble state, and the NI concentration was determined as follows. Aliquots of 200-μL NI-LN suspensions were dissolved and diluted with methanol and then analyzed with high-pressure liquid chromatography (HPLC, LC-20AT; Shimadzu, Kyoto, Japan). The analytical column was a three-μm Cadenza CD-C18 (4.6 mm × 150 mm, Imtakt Corp., Kyoto, Japan). The detector used a 236-nm ultraviolet (UV) source, the column temperature was 40 °C, the mobile phase was methanol/water, 0.2% formic acid (60/40, *v/v*), 60–90% methanol/10 min, the flow rate was 0.45 mL/min, and the injection volume was two µL. For PHT concentrations, the UV source was 213 nm, the column temperature was 40 °C, the mobile phase was methanol/water, 0.2% formic acid (70/30, *v/v*), 70–95% methanol/10 min, the flow rate was 0.40 mL/min, and the injection volume was five µL. 

The drug entrapment efficiency (EE) of the LNs was determined by the amount of free drug after ultrafiltration. Drug–LN suspensions (500-μL) were placed on an ultrafilter in a centrifuge tube (Amicon^®^ Ultra-0.5 Centrifugal Filter Devices, 10 K device 10,000 MNWL; Merck Millipore Ltd., Billerica, MA, USA) and centrifuged at 10,000 rpm at 4 °C for 10 min. This was repeated after 500 µL of water was added. The ultrafiltrate containing the free drug was analyzed by HPLC. The entrapment efficiency was thus: Entrapment efficiency (%) = (total drug content − free drug content)/total drug content × 100

### 2.6. Structural Analysis of Lipid Nanoparticles

Cryo-TEM images were acquired on a JEM-2100F microscope (JEOL Co., Ltd., Tokyo, Japan). Hydrophilic treatment of a 200-mesh copper grid covered with a perforated polymer film (Nisshin EM Co. Ltd., Tokyo, Japan) was performed for 60 s using an HDT-400 device (JEOL Co., Ltd. Tokyo, Japan). A two-μL aliquot of each lipid nanoparticle suspension was then applied to the hydrophilic grid. The grid was then blotted with filter paper for three seconds and immediately vitrified in liquid ethane cooled with liquid nitrogen, using a Leica EM CPC cryofixation system (Leica Microsystems GmbH, Wetzlar, Germany). Frozen samples were maintained at −170 °C, using a Gatan 626 cryo-holder (Gatan, Inc., Pleasanton, CA, USA). The cryo-TEM was operated at 120 kV and a provided a magnification of ×50,000.

AFM imaging in alternating tapping mode was outperformed with an MFP-3D microscope (Asylum Research, Santa Barbara, CA, USA), equipped with a silicon nitride cantilever OMCL-TR400PSA (Olympus Co., Ltd., Tokyo, Japan). Mica modified with 3-(aminopropyl)triethoxysilane (APTES) was prepared by exposing freshly cleaved mica to an APTES atmosphere for one hour at room temperature. This strengthened the weak binding of the lipid nanoparticles to the surface in an aqueous environment. The lipid nanoparticle suspensions were diluted 300 times with deionized distilled water, and then deposited onto the APTES-modified mica surfaces. After incubation for one minute at room temperature, excess nanoparticles were removed by flushing with deionized distilled water. All of the 1024 × 128-pixel AFM images with areas of 2 × 2 μm^2^ were recorded at a scan speed of 0.50 Hz at 20 ± 2 °C. The height and diameter of each lipid nanoparticle was analyzed by the MFP-3D program, written in IGOR-PRO software (Wavemetrics, Portland, OR, USA).

## 3. Results

### 3.1. Stractural Feature of Standard NI-LNs

#### 3.1.1. Cryo-TEM Images of Standard NI-LNs

Cryo-TEM provided images of the suspensions directly without negative staining or drying, and the sample stage could be tilted for imaging at various angles [38]. Figure 1 shows cryo-TEM images of NI-LNs prepared with HSPC/DPPG (5/1) before and after tilting the sample stage by 20 degrees. The shape of most particles changed after the tilting. For example, the rod-like particle before tilting (circled in Figure 1A) became circular (Figure 1B); whereas, the opposite occurred for the particles indicated by arrows. These changes indicated flat particles, as previously reported [39]. There were also small micelles about 10 nm in diameter, as indicated by arrowheads. Thus, flat particles and micelles coexisted in standard NI-LN suspensions.

#### 3.1.2. AFM Images of Standard NI-LNs

AFM provides three-dimensional images. Figure 2A shows AFM images of standard NI-LNs, and Figure 2B exhibits the cross-sectional profile taken along the red line in Figure 2A, where the blue point corresponds to zero in the profile. The NI-LNs were five-nm high and about 50 nm in diameter (Figure 2B), which was consistent with the cryo-TEM images. The thickness of a lipid bilayer is four to six nm [40,41]; thus, the NI-LNs were a single lipid bilayer. Spherical vesicles such as liposomes sometimes strongly adsorbed onto the substrate, resulting in heights over 10 nm that indicated double lipid bilayers [42]. The phospholipids used here were saturated and relatively rigid, and would be difficult to deform during adsorption. Therefore, the images suggested that the standard NI-LNs seem to be disk-like bicelles (Figure 3). 

### 3.2. Effect of Long-Term Storage and Stabilization on LN Structure

The mean particle size and drug concentration without stabilizers did not changed for four months when NI-LNs were stored in a cool, dark place [28]. There was no aggregation of particles or drug leakage, indicating that the NI-LNs had long-term stability. However, it was unclear whether the bicelle structure was maintained, considering that the lipids in the plane and in the edges area were miscible and could coalescence into unilamellar vesicles [43]. However, NI-LNs stored for four months maintained their mean particle size and structure (Figure 4A), and remained dispersed because of the electrostatic repulsion of the negatively-charged DPPG. Hence, the bicelle structure may have increased the LN stability. In addition, cryo-TEM images of NI-LNs that were freeze-dried with sucrose (5 *w*/*v*%) for 48 h and then re-hydrated in distilled water are shown in Figure 4B. The particle size looked slightly bigger after freeze-drying, but the bicelle structure was preserved. Thus, freeze-drying and the addition of a lyophilization stabilizer did not affect the NI-LN structure. 

As for determination of the mean particle size, the following two ways were used; one is by cryo-TEM images (Table 1), and the other is by zetasizer using DLS (Table 2). Especially, because bicelles and micelles coexisted in NI-LNs, analysis of the mean particle size for each particle would be useful by cryo-TEM images, and 100 particles in cryo-TEM images printed on paper were measured with a ruler (Table 1). The ratio of bicelle/micelle in NI-LNs just after preparation was about 8/2, and the mean particle size of bicelles was 26.8 nm. On the other hand, the data by DLS showed the value of total particles, including bicelles and micelles, as 47.5 nm. These results suggested that the mean particle size obtained from the cryo-TEM images (Table 1) was smaller than that obtained from DLS (Table 2). Cryo-TEM and DLS data were based on the number reference and on scattering intensity from Brownian motion, respectively. Previously, it was reported that DLS tends to overestimate the hydrodynamic diameter because large particles strongly influences scattering intensity [44]. In addition, DLS furnished the hydrodynamic diameter depending on the assumption that the particles are spherical [45], but the bicelles in this study were not spherical. Therefore, the difference of measurement principle would be involved in the differences of particle size.

After re-hydration, PDI increased as well as mean particle size (Table 2), so particle size distribution was broader to the bigger value. In addition, drug concentration was decreased to half (30.7 µg/mL) after re-hydration (Table 2). Freeze-drying with cryoprotectant has been reported to cause leakage of the encapsulated drug [46], suggesting that the insertion of sucrose molecules in the phospholipid molecules of the lipid bilayer might be involved in the drug leakage without any significant structural change. Furthermore, the mean particle size was 70.1 nm one month after re-hydration (data not shown), and it exhibited good redispersibility. Therefore, freeze-drying was useful to give long-term stability to LNs.

### 3.3. Effect of Encapsulated Drugs on LN Structure

Next, to determine the effect of encapsulated drugs on the LN structure, LNs without any drugs or with other water-insoluble drugs was prepared. A preliminary study demonstrated that eight water-insoluble drugs encapsulated in LNs (e.g., ibuprofen and indomethacin) improved their solubility (100 µg/mL drugs), while PHT-encapsulated LNs (PHT-LNs) had only a 35 µg/mL drug concentration, suggesting that different structures might be observed, and PHT was chosen as the other drug.

Figure 5 shows cryo-TEM images of LNs (5A) and PHT-LNs (5B) where bicelles and micelles coexisted in both suspensions, with no difference in each structure or mean particle size compared with that of standard NI-LNs (Table 3). Furthermore, AFM images of LNs revealed low heights with respect their size (Figure 5C,D), indicating bicelle structures. The increase in PDI of LNs might result from the increase in the ratio of micelle (Table 3 and Table 4). The PHT-LNs had physicochemical properties that differed most from those of the standard NI-LNs, but they still had bicelle structures. Therefore, the other drug-encapsulated LNs were probably bicelles as well. The PHT-LN preparation replicated the physicochemical properties, indicating lower drug concentrations relative to NI. A lower *EE* meant that the PHT was difficult to encapsulate, and there was much free PHT (Table 4); however, the results here suggested that the structure of the particles was unchanged with or without encapsulated drugs, irrespective of the drugs.

### 3.4. Effect of Lipid Composition Ratio on LN Structure

NI-LNs were prepared with HSPC/DPPG ratios of (5/0), (5/0.5), (5/1), and (0/5), and LNs (no drug loading) was with HSPC/DPPG (5/0), (5/1), and (0/5). Figure 6 shows cryo-TEM images, and Table 5 lists the physicochemical properties of each lipid nanoparticle. The shapes of NI-LNs prepared with HSPC/DPPG (5/0) did not change markedly when the sample stage was rotated by 20 degrees (Figure 6A,B). Thus, the particles were not flat, and could have been polygonal liposomes [47]. Additionally, a week after preparation, slight aggregation occurred because there was little electrostatic repulsion for lack of DPPG. The DLS mean particle size was about 350 nm two weeks after preparation (data not shown), which was more than three times that just after preparation. The NI concentration was low (2.9 µg/mL) (Table 6), because NI might exist in only the lipid bilayer area of liposomes. Furthermore, as for HSPC/DPPG (5/0), the appearance of LNs suspensions just after 100-pass high-pressure homogenization was still cloudy, indicating that it was difficult to make a particle made from only HSPC smaller, so big particles were trapped by a 200-nm filter membrane. In fact, lipid concentration pass through filter membrane was low (data not shown). So, NI concentration would be low. There was no difference in structure for the no-drug LNs prepared with HSPC/DPPG (5/0) (Figure 6C), relative to that for the NI-LNs. They could not be prepared reproducibly, which indicated large S.D. values of their mean particle size, and their zeta potentials were almost neutral because of lacking DPPG (Table 6). NI-LNs prepared with (5/0.5) (Figure 6D) exhibited morphologies that were between liposomes (5/0) and bicelles (5/1) that corresponded to the DPPG ratio, and the ratio of bicelle/micelle/liposome was about 7/2/1 (Table 5), suggesting that because the amount of DPPG was not enough to form bicelles, the liposomes derived from HSPC still existed. NI-LNs prepared with (0/5) (Figure 6E) were only micelles with a few nanometers in size; there were no bicelles or liposomes, indicating that there were no micelles in the lipid nanoparticle suspensions without DPPG. For HSPC/DPPG (0/5) (Figure 6F), there was no difference in the micellar structure between NI-LNs and LNs. Therefore, the encapsulated drugs did not affect the particle structures. Although big differences in the mean particle size of HSPC/DPPG (0/5) between Table 5 and Table 6 were seen, large particles were not observed in cryo-TEM images, and the value in Table 5 might be more accurate, because DLS is commonly influenced by fewer large particles. Moreover, the highest NI concentration (98.0 µg/mL) (Table 6) was reduced almost by half three weeks after preparation (data not shown), and crystalline NI was observed, suggesting an instability of micelles. Therefore, standard NI-LNs had high NI concentrations and good stability. These results suggested that the lipid composition ratio dramatically affected the structure, the physicochemical properties, and the stability of lipid nanoparticles.

### 3.5. Effect of Phosphatidylcholine Acyl Chain Length on LN Structure

HSPC is a mixture of phosphatidylcholines with different acyl chain lengths (C12–20). Thus, to assess chain-length effects, DPPG (C16) was used as a phosphatidylglycerol and the phosphatidylcholine (PC) chain length was varied. NI-LNs using DMPC (C14), DPPC (C16), and DSPC (C18) were prepared with the ratio PC/DPPG (5/1). In Figure 7, cryo-TEM images of each NI-LN showed that the structure of all the particles made from each PC were bicelles, although the particle sizes of the bicelles were different, while that of the micelles was same between all of the PCs (Table 7). Hence, the PC acyl chain length did not affect the bicelle structure through just changing the particle sizes. 

NI-LNs made with DMPC had substantially smaller mean particle size and bigger PDI relative to other the particles (Table 8). Previously, it was reported that the size distribution had bimodal peaks, and that NI crystal precipitated one month after preparation [48]; thus, NI-LNs using DMPC had different properties from the other particles. Longer alkyl chain lengths result in higher phase transition temperatures and stronger van der Waals interactions. Therefore, the pressure-induced unstable lipid bilayers (electrostatic repulsion from polar head groups) made it difficult to form stable particles with DMPC. 

### 3.6. Effect of High-Pressure Homogenization on LN Structure

Cryo-TEM images in Figure 8 show NI-LNs prepared under different high-pressure homogenization conditions. For a 100-MPa/100 pass under reduced pressure, there were bicelles and micelles similar to standard NI-LNs, but some particles remained large because of the weaker applied shearing force (Figure 8A, Table 9). There were variations that suggested Ostwald ripening. In this case, the stability was inferior to that of NI-LNs prepared under standard conditions. Low NI concentration (Table 10) resulted from the particles having a large size, which were removed by the 200-nm filter membrane. For a 175 MPa/150 pass under increased the pass number, the size became smaller than that of NI-LNs prepared under standard conditions (175 MPa/100 pass). It was possible to observe bicelles that had different widths (Figure 8B). When the pass number increased, the particles had smaller absolute value of zeta potentials and low NI concentration (Table 10). This might be due to the adhesion of the sample (especially DPPG) to the flow path of Microfluidizer^®^. Therefore, the standard condition was the most efficient method. These results suggested that the high-pressure homogenization conditions did not affect the structure, but did affect the size and entrapped drug concentration of the lipid nanoparticles. 

### 3.7. Effect of Cholesterol on LN Structure

As discussed above, lipid nanoparticles prepared with HSPC/DPPG (5/0) were polygonal and thought to be liposomes (Figure 5A,E). Cholesterol (Chol) inserts into PC liposomes and increases the packing between the lipids, which decreases membrane permeability [49]. Some liposomes with Chol had smoother and rounder surfaces relative to those without Chol [47]. Here, Chol was added to NI-LNs prepared with HSPC/DPPG (5/0) to form liposomes, and it was added to standard NI-LNs to compare structural and physicochemical properties. 

Cryo-TEM images in Figure 9 revealed NI-LNs prepared from HSPC/DPPG/Chol (5/0/0) and (5/1/0), as well as from HSPC/DPPG/Chol (5/0/2) and (5/1/2). For HSPC/DPPG/Chol (5/0/2), the surface asperities disappeared, and the particles became smooth (Figure 9B). Since there was no difference in the widths of identical particles before and after sample stage rotation, the structures were spherical. Thus, lipid nanoparticles prepared from HSPC/DPPG/Chol (5/0/0) were liposomes. NI-LNs prepared with HSPC/DPPG/Chol (5/1/2) were bicelles and micelles, similar to standard NI-LNs, but liposomes were also present (Figure 9D). It was reported that when the Chol content was 20 mol% or less in liposomes, there were two phases: a miscible phase composed of Chol and lipids, and a lipid phase composed only of lipids [50]. For a Chol concentration of more than 20 mol%, there were spherical liposomes formed by the miscible phase, and micelles and bicelles formed by the lipid phase. 

As for physicochemical properties (Table 11 and Table 12), NI-LNs prepared with (5/0/0) had almost neutral potential, but the zeta potential increased when prepared with (5/0/2) (Table 12). Furthermore, NI-LNs prepared with (5/1/2) had negative charge, so DPPG had a larger contribution to zeta potential than Chol. NI-LNs prepared with (5/1/2) had bigger PDI values because of the mixture of three structures (Table 11 and Table 12). Moreover, NI-LNs with Chol had increased mean particle sizes and slightly higher NI concentrations (Table 12). Standard NI-LNs stayed dispersed for four months, while aggregation was observed in NI-LNs with Chol two months after preparation (data not shown). This suggested that fluidity was enhanced and that particle-to-particle adsorptions were easier with Chol in the membrane. Robust and stable lipid membranes could be formed without Chol, and NI-LNs had highly stable *EE* values. This was because the difference in curvature in inside and outside small unilamellar vesicles disordered the lipid molecules and affected their permeability. However, the edge curvature in bicelles was large, while the plane curvature was almost zero. The lipids in the latter were packed by high pressure and shearing forces during preparation. Therefore, NI-LNs had good properties without Chol. 

## 4. Discussion

NI-LNs had shown improved solubility, high *EE*, long-term stability, and good redispersibility. To the best of our knowledge, other lipid nanoparticles having these all properties have not been reported yet, suggesting the possibility that NI-LNs might have unique structural features from other lipid nanoparticles. In this study, the structure of the particles was focused on, and cryo-TEM and AFM analyses were performed. From a series of experiments, the structure of NI-LNs was found to be bicelles.

In bicelles, long-chain lipids (C16–20) generally form the planar area, and short-chain lipids (C6–8) form the edge area [51,52]. Diheptanoylphosphatidylcholine (DHPC, C6) is usually used as short-chain lipids forming the edge area, but there are some examples except for short-chain lipids; e.g., dimyristoylphosphoethanolamine (C14) introduced by lanthanide to the head group [53], n-dodecylphosphocholine (C12) that had one alkyl chain [54], and ceramide (C16) bound to polyethylene glycol (PEG)-5000 [55]. As the feature of these materials, they had large head areas. It was reported that hydrophilic surfaces with high curvature can be formed by anionic lipids because the head volume is expanded by strong electrostatic repulsion in low ionic strength media [56]. In the present study, the bicelles formed here used phospholipids with similar alkyl chain lengths: HSPC (C12–20) and DPPG (C16). However, anionic DPPG might have a large head area, because there were micelles when only DPPG was contained in the formulation in cryo-TEM images and PG with a negatively charged head group could be self-assembled at high curvatures. Therefore, we speculated that DPPG was localized in the edge area, and HSPC was in the flat area to form bicelles (Figure 3). 

It was reported that spherical liposomes were prepared using PC and PG by other methods [57,58]. The general preparation method of bicelles [54,59] is quite different from our method using the roll mill and microfluidizer. Nevertheless, the disk structure was formed in this work, and this may be explained by the following two reasons: one is a high shearing force, and the other is heating by microfluidization. First, the drugs, PC and PG, were miscible by roll milling, and liposome-like particles assembled in distilled water during pre-dispersion were collapsed by the high shearing force during microfluidizer miniaturization. The PG was subsequently assembled along the edge as a cap for the flat lipid bilayer, forming the disk-like particle. In addition, it was reported that thermosensitive liposomes containing micelle-forming components such as distearoylphosphoethanolamine-PEG2000 or lysolipids changed to bilayer discs after a repeated cycling through the phase transition temperature (T_c_) [60]. Therefore, the spherical particles may have changed to be bicelles, because of the heating caused by the miniaturization process.

In addition to temperature, the determining factors for whether lipid assembly forms bicelle are lipid concentration and the molar ratio of two phospholipids. Lipid concentration (20% or more) is necessary to form bicelles. It has been reported that when the DMPC/DHPC bicellar system was diluted to 0.07%, the disk form changed to be vesicle [61]. This transformation mechanism was that DHPC, having a relatively high solubility, was separated from the edge of the disk; then, some bilayers integrated, and the size became bigger, resulting in the growing and unstable bilayers transform vesicle. To protect this transformation of bicelles under dilute conditions, bicelle-encapsulated liposomes named “bicosomes” were developed [62]. However, our bicelles can form disk structures at low lipid concentration (0.5% lipid concentration) and be stable for four months without any significant change of the structure (Figure 4A). We speculated that due to the strong hydrophobic interaction between HSPC and the longer acyl chain of DPPG in this work, DPPG might strongly cap the lipid bilayer of HSPC and prevent the leakage of encapsulated drug, giving the relatively long-term stability, contrary to previously reported DMPC/DHPC bicelles. In addition, the concentrated bicelles by centrifugation can also maintain their disk structure, suggesting that our methodology can prepare bicelles with arbitrary concentrations. Furthermore, it is suggested that our bicelles can maintain their disk structure even when intravitally administrated in vivo, where a high amount of liquid exists. 

As for the molar ratio of two phospholipids, when the ratio of long-chain to short-chain (the ratio q) fatty acids is 2.5 or less, the bicelles are isotropic; while, when the ratio of q is 3.0 or more, the bicelles are anisotropic in the magnetic field [63], indicating that the particle size of bicelles is small and big, respectively. In our case, when assumed with HSPC as the long chain and DPPG as the short chain, the ratio of q made with HSPC/DPPG (5/1, for standard) and (5/0.5) is 5.0 and 10, respectively, and no significant differences in mean particle size between them are seen (Table 6). Therefore, when the ratio of q is changed to be smaller, bicelles with smaller mean particle size would be obtained. There are some reports about bicelles for the DDS field. For example, Barbosa-Barros reported that bicelles have been used for drug penetration carrier through skin [64]. In addition, organic–inorganic hybrid bicelles named “cerasomes” were also developed [65]. These hybrid bicelles were reported to show a unique drug release profile under an acidic pH environment. However, the amount of published literatures is small. For our bicelles, although the drug release study, membrane permeation study using caco-2, in vivo biocompatibility, and pharmacokinetic study should be determined, further applications using our stable bicelles are also expected for the DDS field. In addition, the differences between our bicelles and other bicelles prepared with general materials using general methods would be necessary to compare in the future.

## 5. Conclusions

Roll-mill grinding and high-pressure homogenization were used to prepare lipid nanoparticles from a neutral PC lipid and a negatively charged PG lipid under various conditions. Standard NI-LNs with HSPC/DPPG (5/1) were disk-like bicelles. No changes were observed after long-term storage, and NI-LNs freeze-dried with sucrose and re-hydration. Liposomes were formed for HSPC/DPPG (5/0), and micelles were formed for HSPC/DPPG (0/5). The addition of Chol created smooth particle surfaces for HSPC/DPPG (5/0/2). A serious of experiments and results demonstrated that the standard NI-LN bicelles have the best physicochemical properties, and further DDS study would be desired using these stable bicelles.

## Figures and Tables

**Figure 1 nanomaterials-08-00998-f001:**
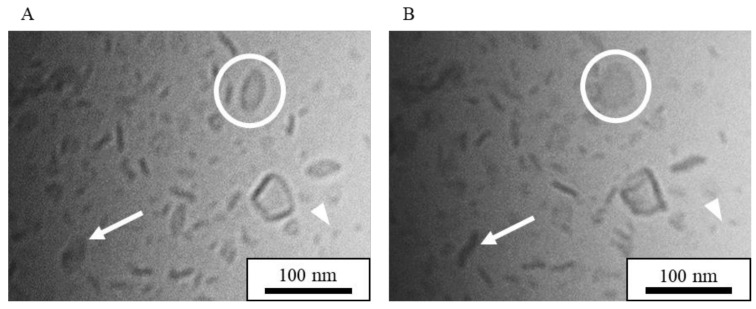
Cryo transmission electron microscopy (cryo-TEM) images of standard nifedipine-encapsulated lipid-based nanoparticles (NI-LNs) prepared with hydrogenated soybean phosphatidylcholine (HSPC) and dipalmitoylphosphatidylglycerol (DPPG) (5/1). (**A**) Before sample stage tilting, and (**B**) after tilting 20 degrees.

**Figure 2 nanomaterials-08-00998-f002:**
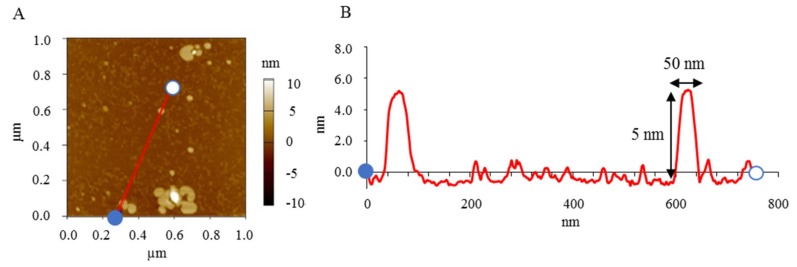
Atomic force microscopy (AFM) imaging of standard NI-LNs. (**A**) AFM image and (**B**) cross-sectional profile.

**Figure 3 nanomaterials-08-00998-f003:**
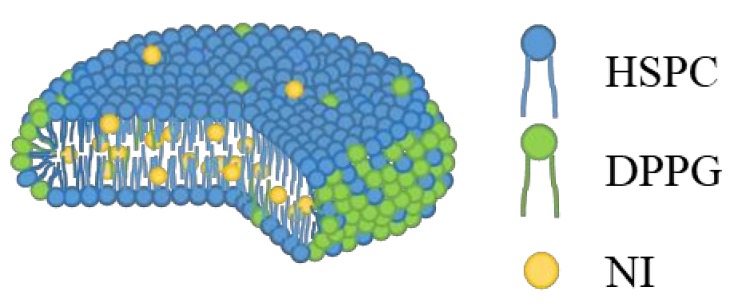
Schematic of an NI-LN bicelle.

**Figure 4 nanomaterials-08-00998-f004:**
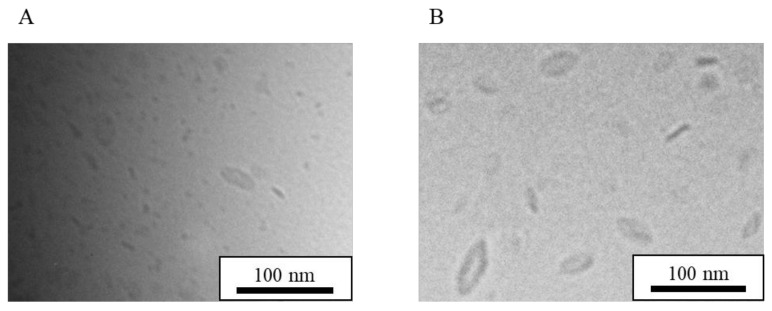
Cryo-TEM images of NI-LNs (**A**) stored for four months in a cold, dark place, and (**B**) re-hydrated after freeze-drying with sucrose.

**Figure 5 nanomaterials-08-00998-f005:**
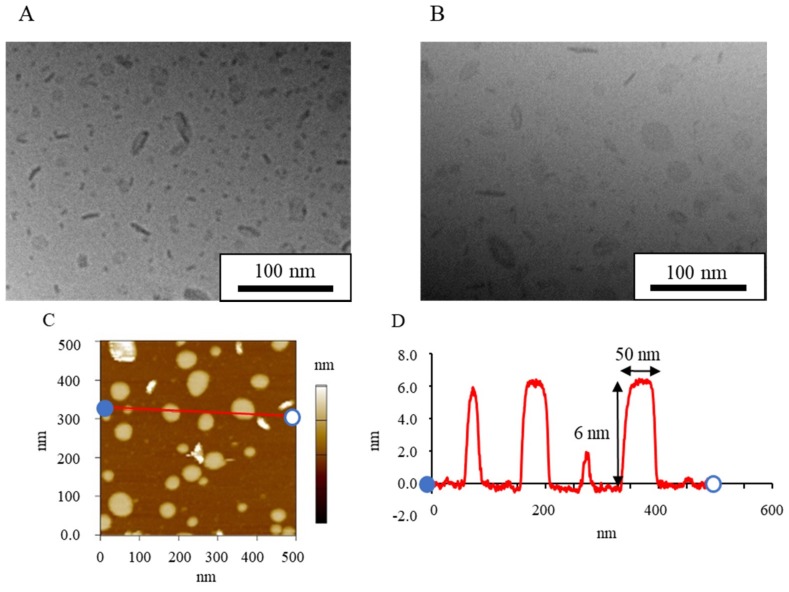
Structural analysis of LNs and phenytoin (PHT)-LNs. Cryo-TEM image of (**A**) LNs and (**B**) PHT-LNs. (**C**) AFM image of LNs and (**D**) the cross-sectional profile.

**Figure 6 nanomaterials-08-00998-f006:**
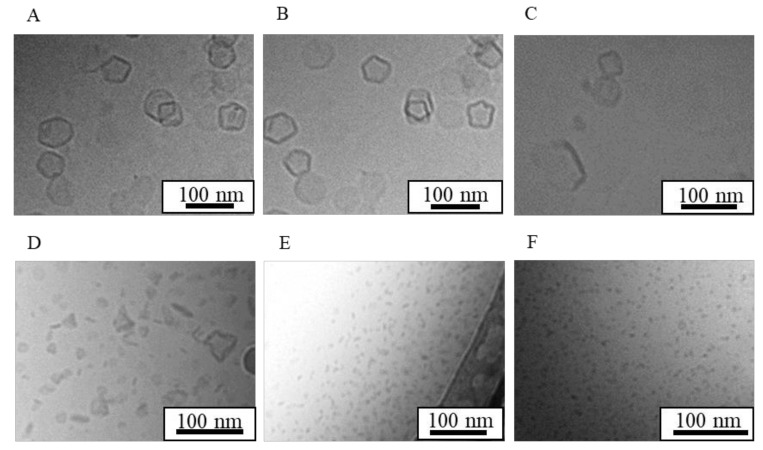
Cryo-TEM images of NI-LNs and LNs structures made with different HSPC/DPPG ratios. HSPC/DPPG (5/0) NI-LNs with the sample stage tilted 20 degrees (**A**) before and (**B**) after. (**C**) HSPC/DPPG (5/0) LNs. NI-LNs made from (**D**) HSPC/DPPG (5/0.5) and (**E**) HSPC/DPPG (0/5). (**F**) HSPC/DPPG (0/5) LNs.

**Figure 7 nanomaterials-08-00998-f007:**
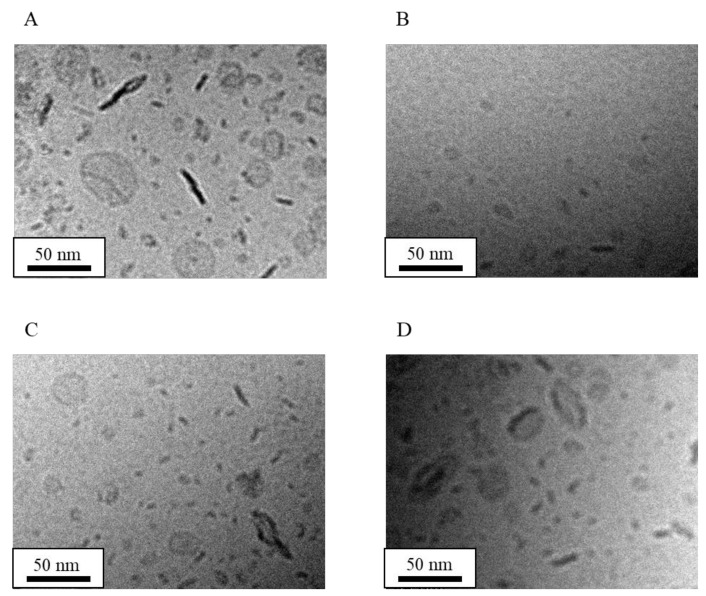
Cryo-TEM images of NI-LN structures having various alkyl chain lengths. (**A**) HSPC/DPPG (5/1), (**B**) 1,2-dimyristoyl-sn-glycero-3-phosphocholine (DMPC)/DPPG (5/1), (**C**) 1,2-dipalmitoyl-sn-glycero-3-phosphocholine (DPPC)/DPPG (5/1) and (**D**) 1,2-distearoyl-sn-glycero-3-phosphocholine (DSPC)/DPPG (5/1).

**Figure 8 nanomaterials-08-00998-f008:**
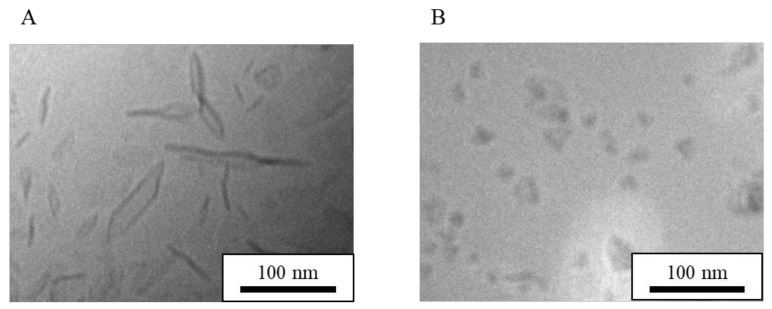
Cryo-TEM images of NI-LN structures made from HSPC/DPPG (5/1) under different high-pressure homogenization conditions. (**A**) 100 MPa/100 pass and (**B**) 175 MPa/150 pass.

**Figure 9 nanomaterials-08-00998-f009:**
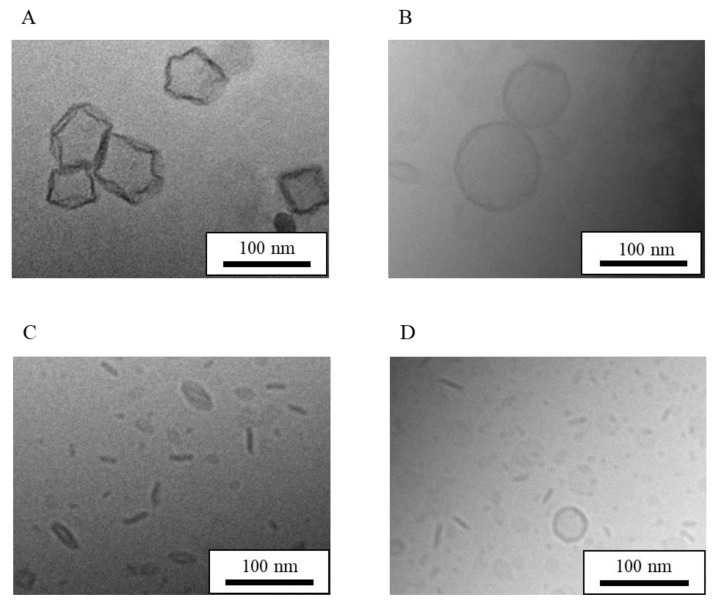
Cryo-TEM images of NI-LNs structures with/without cholesterol (Chol). (**A**) HSPC/DPPG/Chol (5/0/0), (**B**) HSPC/DPPG/Chol (5/0/2), (**C**) HSPC/DPPG/Chol (5/1/0) and (**D**) HSPC/DPPG/Chol (5/1/2).

**Table 1 nanomaterials-08-00998-t001:** Effect of long-term storage and freeze-drying on mean particle size measured by cryo-TEM images.

Formulation	Bicelle (nm)	Micelle (nm)	Mean Particle Size (nm)
Just after preparation	26.8 ± 8.4 (*n* = 81)	8.0 ± 1.7 (*n* = 19)	23.2 ± 10.6 (*n* = 100)
Stored for four months at a cold dark place	23.4 ± 10.9 (*n* = 59)	7.6 ± 1.8 (*n* = 41)	17.0 ± 11.5 (*n* = 100)
Re-hydrated after freeze-drying with sucrose	39.7 ± 12.1 (*n* = 91)	11.1 ± 1.0 (*n* = 9)	37.1 ± 14.2 (*n* = 100)

Data obtained by cryo-TEM images are average values of particle length (mean ± S.D.).

**Table 2 nanomaterials-08-00998-t002:** Effect of long-term storage and freeze-drying on physiochemical properties of NI-LNs by dynamic light scattering (DLS) and high-pressure liquid chromatography (HPLC) measurements. EE: entrapment efficiency, PDI: polydispersity index.

Formulation	Mean Particle Size (nm)	PDI	Zeta Potential (mV)	Drug Concentration (µg/mL)	*EE* (%)
Just after preparation	47.5 ± 2.9	0.307 ± 0.027	−46.8 ± 9.6	67.3 ± 11.3	97.1 ± 1.6
Stored for four months at a cold dark place	46.3 ± 1.0	0.310 ± 0.033	−52.1 ± 2.8	67.4 ± 6.2	98.1 ± 0.3
Re-hydrated after freeze-drying with sucrose	63.7 ± 1.2	0.351 ± 0.007	−47.0 ± 4.1	30.7 ± 7.6	97.8 ± 0.5

**Table 3 nanomaterials-08-00998-t003:** Effect of encapsulated drug on mean particle size measured by cryo-TEM images.

Model Drug	Bicelle (nm)	Micelle (nm)	Mean Particle Size (nm)
NI	26.8 ± 8.4 (*n* = 81)	8.0 ± 1.7 (*n* = 19)	23.2 ± 10.6 (*n* = 100)
None	21.1 ± 5.6 (*n* = 68)	6.1 ± 2.1 (*n* = 32)	16.3 ± 8.7 (*n* = 100)
PHT	26.5 ± 9.3 (*n* = 81)	8.1 ± 1.6 (*n* = 19)	23.0 ± 11.1 (*n* = 100)

Data obtained by cryo-TEM images are average values of particle length (mean ± S.D.).

**Table 4 nanomaterials-08-00998-t004:** Effect of encapsulated drug on physiochemical properties of LNs by DLS and HPLC measurements.

Model Drug	Mean Particle Size (nm)	PDI	Zeta Potential (mV)	Drug Concentration (µg/mL)	*EE*(%)
NI	47.5 ± 2.9	0.307 ± 0.027	−46.8 ± 9.6	67.3 ± 11.3	97.1 ± 1.6
None	43.3 ± 0.7	0.357 ± 0.086	−55.6 ± 4.2	-	-
PHT	46.0 ± 4.4	0.267 ± 0.003	−30.7 ± 6.4	30.5 ± 2.8	59.4 ± 7.6

**Table 5 nanomaterials-08-00998-t005:** Effect of HSPC/DPPG molar ratio on mean particle size measured by cryo-TEM images.

HSPC/DPPG (Molar Ratio)		Bicelle (nm)	Micelle (nm)	Liposome (nm)	Mean Particle Size (nm)
NI-LNs	(5/0)	-	-	77.4 ± 31.5 (*n* = 100)	77.4 ± 31.5 (*n* = 100)
	(5/0.5)	24.1 ± 9.4 (*n* = 71)	9.0 ± 2.1 (*n* = 22)	43.9 ± 10.1 (*n* = 7)	22.2 ± 12.0 (*n* = 100)
	(5/1)	26.8 ± 8.4 (*n* = 81)	8.0 ± 1.7 (*n* = 19)	-	23.2 ± 10.6 (*n* = 100)
	(0/5)	-	8.6 ± 2.4 (*n* = 100)	-	8.6 ± 2.4 (*n* = 100)
LNs	(5/0)	-	-	80.5 ± 40.6 (*n* = 20)	80.5 ± 40.6 (*n* = 20)
	(5/1)	21.1 ± 5.6 (*n* = 68)	6.1 ± 2.1 (*n* = 32)	-	16.3 ± 8.7 (*n* = 100)
	(0/5)	-	10.9 ± 1.7 (*n* = 100)	-	10.9 ± 1.7 (*n* = 100)

Data obtained by cryo-TEM images are average values of particle length (mean ± S.D.).

**Table 6 nanomaterials-08-00998-t006:** Effect of HSPC/DPPG molar ratio on physiochemical properties on LNs by DLS and HPLC measurements. PDI: polydispersity index.

HSPC/DPPG (Molar Ratio)		Mean Particle Size (nm)	PDI	Zeta Potential (mV)	Drug Concentration (µg/mL)	*EE* (%)
NI-LNs	(5/0)	93.1 ± 102.4	0.443 ± 0.049	−5.5 ± 9.1	2.9 ± 2.5	63.4 ± 55.1
	(5/0.5)	50.7 ± 3.1	0.294 ± 0.025	−43.6 ± 7.9	58.0 ± 25.8	97.0 ± 1.8
	(5/1)	47.5 ± 2.9	0.307 ± 0.027	−46.8 ± 9.6	67.3 ± 11.3	97.1 ± 1.6
	(0/5)	54.5 ± 28.4	0.793 ± 0.210	−49.9 ± 3.7	98.0 ± 24.2	99.3 ± 0.1
LNs	(5/0)	217.7 ± 135.1	0.238 ± 0.026	−13.5 ± 0.4	-	-
	(5/1)	48.7 ± 9.4	0.357 ± 0.086	−55.6 ± 4.2	-	-
	(0/5)	92.4 ± 10.3	0.540 ± 0.029	−55.4 ± 5.4	-	-

**Table 7 nanomaterials-08-00998-t007:** Effect of different alkyl chain lengths on mean particle size measured by cryo-TEM images.

Formulation	Bicelle (nm)	Micelle (nm)	Mean Particle Size (nm)
HSPC/DPPG (5/1)	26.8 ± 8.4 (*n* = 81)	8.0 ± 1.7 (*n* = 19)	23.2 ± 10.6 (*n* = 100)
DMPC/DPPG (5/1)	15.6 ± 3.3 (*n* = 40)	6.7 ± 1.0 (*n* = 15)	13.2 ± 4.9 (*n* = 55)
DPPC/DPPG (5/1)	25.7 ± 11.3 (*n* = 82)	7.2 ± 1.4 (*n* = 18)	22.4 ± 12.5 (*n* = 100)
DSPC/DPPG (5/1)	26.4 ± 8.8 (*n* = 77)	6.1 ± 0.6 (*n* = 23)	21.7 ± 11.5 (*n* = 100)

Data obtained by TEM images are average values of particle length (mean ± S.D.).

**Table 8 nanomaterials-08-00998-t008:** Effect of different alkyl chain lengths on physiochemical properties of NI-LNs by DLS and HPLC measurements.

Formulation	Mean Particle Size (nm)	PDI	Zeta Potential (mV)	Drug Concentration (µg/mL)	*EE* (%)
HSPC/DPPG (5/1)	47.5 ± 2.9	0.307 ± 0.027	−46.8 ± 9.6	67.3 ± 11.3	97.1 ± 1.6
DMPC/DPPG (5/1)	34.4 ± 17.2	0.668 ± 0.194	−35.1 ± 10.0	61.0 ± 19.0	98.9 ± 0.2
DPPC/DPPG (5/1)	50.4 ± 11.1	0.361 ± 0.086	−53.1 ± 1.8	61.5 ± 19.6	97.9 ± 1.9
DSPC/DPPG (5/1)	62.5 ± 5.9	0.285 ± 0.041	−47.6 ± 3.7	59.1 ± 17.9	97.7 ± 2.3

**Table 9 nanomaterials-08-00998-t009:** Effect of different conditions of high-pressure homogenization on mean particle size measured by cryo-TEM images.

Conditions	Bicelle (nm)	Micelle (nm)	Mean Particle Size (nm)
175 MPa/100 pass	26.8 ± 8.4 (*n* = 81)	8.0 ± 1.7 (*n* = 19)	23.2 ± 10.6 (*n* = 100)
100 MPa/100 pass	47.0 ± 21.5 (*n* = 84)	9.4 ± 1.3 (*n* = 16)	41.0 ± 24.0 (*n* = 100)
175 MPa/150 pass	23.1 ± 6.0 (*n* = 49)	10.4 ± 1.9 (*n* = 11)	20.7 ± 7.3 (*n* = 60)

Data obtained by cryo-TEM images are average values of particle length (mean ± S.D.).

**Table 10 nanomaterials-08-00998-t010:** Effect of different conditions of high-pressure homogenization on physiochemical properties of NI-LNs by DLS and HPLC measurements.

Conditions	Mean Particle Size (nm)	PDI	Zeta Potential (mV)	Drug Concentration (µg/mL)	*EE* (%)
175 MPa/100 pass	47.5 ± 2.9	0.307 ± 0.027	−46.8 ± 9.6	67.3 ± 11.3	97.1 ± 1.6
100 MPa/100 pass	67.9 ± 8.5	0.251 ± 0.004	−41.2 ± 19.7	24.6 ± 6.7	98.7 ± 0.6
175 MPa/150 pass	39.1 ± 14.4	0.264 ± 0.004	−28.1 ± 31.1	37.0 ± 3.3	99.2 ± 0.4

**Table 11 nanomaterials-08-00998-t011:** Effect of cholesterol on mean particle size measured by cryo-TEM images.

Formulation	Bicelle (nm)	Micelle (nm)	Liposome (nm)	Mean Particle Size (nm)
HSPC/DPPG/Chol (5/0/0)	-	-	77.4 ± 31.5 (*n* = 100)	77.4 ± 31.5 (*n* = 100)
HSPC/DPPG/Chol (5/0/2)	-	-	46.3 ± 9.6 (*n* = 100)	46.3 ± 9.6 (*n* = 100)
HSPC/DPPG/Chol (5/1/0)	26.8 ± 8.4 (*n* = 81)	8.0 ± 1.7 (*n* = 19)	-	23.2 ± 10.6 (*n* = 100)
HSPC/DPPG/Chol (5/1/2)	22.8 ± 8.2 (*n* = 60)	7.6 ± 1.0 (*n* = 32)	34.7 ± 8.9 (*n* = 8)	25.8 ± 8.8 (*n* = 100)

Data obtained by cryo-TEM images are average values of particle length (mean ± S.D.).

**Table 12 nanomaterials-08-00998-t012:** Effect of cholesterol on physiochemical properties of LNs by DLS and HPLC measurements.

HSPC/DPPG/Chol	Mean Particle Size (nm)	PDI	Zeta Potential (mV)	Drug Concentration (µg/mL)	*EE* (%)
(5/0/0)	93.1 ± 102.4	0.443 ± 0.049	−5.5 ± 9.1	2.9 ± 2.5	63.4 ± 55.1
(5/0/2)	69.2 ± 6.2	0.368 ± 0.004	44.0 ± 1.6	12.6 ± 1.0	98.8 ± 0.6
(5/1/0)	47.5 ± 2.9	0.307 ± 0.027	−46.8 ± 9.6	67.3 ± 11.3	97.1 ± 1.6
(5/1/2)	80.0 ± 21.3	0.468 ± 0.004	−55.2 ± 1.1	109.2 ± 17.1	98.2 ± 1.5

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
