# Peer review of "Combination of Roll Grinding and High-Pressure Homogenization Can Prepare Stable Bicelles for Drug Delivery"

_nanomaterials, 2018, doi:10.3390/nano8120998_

Reviewer 1 Report

        The manuscript describes the physico-chemical characterization of various bicelles containing nifedipine made up of different lipid mixture and obtained by roll grinding and high-pressure homogenization. The rationale was provided and the experiments supported the thesis provided by the authors. The following criticisms should be properly addressed in order to improve the quality of the manuscript.

        Page 3, line 117 – “The NI-LN suspension was filtered with a 0.2-μm polytetrafluoroethylene or  cellulose acetate membrane filter….” – More details are required (equipment, for example syringe or extruder, cycles, etc). Moreover, the authors did not describe any purification methods (centrifugation, dialysis, etc). Have been the non-encapsulated drugs separated by the systems? It could be detrimental for the EE% analysis.

        Page 3, line 120 – “2.3. Freeze-drying and reconstitution of NI-lipid nanoparticles” The authors should justify the choice of sucrose instead of other cryoprotectants (i.e. mannose, glucose, trehalose, etc) and the concentration used.

        Page 3, line 126 – “2.4. Measurement of mean particle size and zeta potential” – the dilution used should be added.

        Page 6, line 214 – “On the other hand, the data by DLS shows the value of total particles including bicelles and micelles and was 47.5 nm”. The authors should evaluate the data as number(%), volume(%) and intensity(%) in order to furnish a plausible size distribution of the various systems.          

        The modulation of the physico-chemical parameters of the various formulations should be evaluated in saline (0.9% NaCl).

        Tables 5 and 6 – the type of ratio (molar) should be reported.

        The quality of Figure 8 panel B must be improved.

        Author Response

        Response to Reviewers' Comments

   We thank the reviewers for the very valuable comments. We have addressed all the issues raised by the reviewers in a point-by-point manner as noted below. The changes made in the manuscript have been marked in red.

        Reviewer #1. We thank the reviewer for his or her useful comments.

        Response:

Comments

Our response

1

Page 3, line 117 – “The NI-LN suspension was   filtered with a 0.2-μm polytetrafluoroethylene or cellulose acetate membrane   filter….” – More details are required (equipment, for example syringe or   extruder, cycles, etc). Moreover, the authors did not describe any   purification methods (centrifugation, dialysis, etc). Have been the   non-encapsulated drugs separated by the systems? It could be detrimental for   the EE% analysis.

   We are sorry for insufficient explanation. After high-pressure homogenization with a pass cycle of 100, the filtration was carried out by suction aspirator using a 0.2-µm cellulose acetate membrane filter (Toyo Roshi Kaisha Ltd.) to remove the particles whose size was over 200 nm. Based on the reviewer’s comment, the following sentence has been revised in the revised manuscript.

         (page 3, line 119)

: The NI-LN suspension was filtered by suction aspirator with a 0.2-µm  polytetrafluoroethylene or cellulose acetate membrane filter (Toyo Roshi   Kaisha Ltd.) to remove the particles whose size was over 200 nm.

 In addition,  after filtration, we did not perform any purification methods like centrifugation and dialysis. It means that the sample solution contains NI-encapsulated LNs which can be pass through the 0.2-μm filter and non-encapsulated drugs. However, when measuring the Entrapment efficiency (EE)(%) after ultrafiltration by ultrafilter, EE values were over 97% in Table 2   (original manuscript) and free drug percentage (%) was just 2-3%. Therefore, since the free drug percentage was quite low, we think additional purification methods are not necessary in this study.Thank you for your comments.   

2

Page 3, line 120 – “2.3. Freeze-drying   and reconstitution of NI-lipid nanoparticles” The authors should justify the   choice of sucrose instead of other cryoprotectants (i.e. mannose, glucose,   trehalose, etc) and the concentration used.

        Thank you for your comments. Actually, we previously reported that when adding some   types of cryoprotectants (glucose, fructose, maltose or sucrose) at  various concentrations to NI-LNs and then freeze-drying,  good re-dispersibility of NI-LNs with disaccharides (maltose and sucrose) were shown compared with monosaccharides (glucose and sucrose)(Ohshima  H, et al. (2009), Ref #28 in original manuscript). In addition, Barman et al   also reported that in comparison to monosaccharide, disaccharide showed higher cryoprotectant activity when adding at a concentration of 5% (w/v). Therefore,   we used sucrose as one of good cryoprotectants in this study.

Based on the reviewer’s comment, the following sentence has been revised and new reference has been added as #37 in the revised manuscript.

        (Page 3, line 121)

        : Freeze-drying was performed with 2 ml of the NI-LN suspensions placed in a vial containing 100 mg of sucrose   (5%, w/v) as a cryoprotectant. Sucrose was selected based   on the previous reports [28][37].

        (Reference)

37. Barman, R.; Iwao, Y.; Funakoshi, Y.;   Ranneh, A.; Noguchi, S.; Wahed, M.; Itai, S. Development of Highly Stable   Nifedipine Solid–Lipid Nanoparticles. Chem.   Pharm. Bull. 2011, 47, 4691–4693.

3

Page 3, line 126 – “2.4. Measurement   of mean particle size and zeta potential” – the dilution used should be   added.

        When measuring mean particle size, PDI and zeta   potential by zetasizer, the values of mean count rate of intact samples were 200–500 kcps which are suitable for this measurement. Therefore, we did not perform the dilution in this study.

Based on the reviewer’s comment, the following sentence has been added in the revised   manuscript.

        (Page 3, line 133)  

        : The mean particle size and the   zeta potential of LN suspensions were measured with dynamic light scattering   (DLS) (Zetasizer nano ZS, Malvern Instruments Ltd., Worcestershire, UK) at a  scattering angle of 90° at room temperature (25°C). Mean particle sizes were   based on the scattering intensity, while the zeta potential was based on   electrophoretic mobility. Since the values of mean count rate of the samples were 200–500 kcps which are suitable for this   measurement, intact samples without dilution were used in this measurement.  We repeated measurement three times per a sample. 

4

Page 6, line 214 – “On the other hand,   the data by DLS shows the value of total particles including bicelles and   micelles and was 47.5 nm”. The authors should evaluate the data as number (%),   volume (%) and intensity (%) in order to furnish a plausible size   distribution of the various systems.

         Thank you for your comment. We agree with your   idea and we always have to consider which analyses (intensity-based, Volume-based   and Number-based) by zetasizer would be appropriate for measuring   nanoparticles. The values of mean particle size and PDI were determined by scattering   intensity, as described in line 129 of original manuscript. Definitely, since   our LNs has some types of particles, like bicelles, micelles and liposomes, number-based   analyses would be appropriate if we show the data of the particle size distribution. However, in this study, we   want to just determine the mean particle size of LNs, and even if the   intensity-based analysis was changed to volume-based and number-based   analyses, all data of mean particle sizes (z-values) of NI-LNs were same   because cumulant method was applied in all analyses. In addition, all the   data of quality report were “good”, meaning this analysis was acceptable. Taken   together, in this study, we want to show only the data of mean particle size   and we think intensity-based analysis is enough. We hope the reviewer   understanding our thought.

5

The modulation of the physico-chemical   parameters of the various formulations should be evaluated in saline (0.9%   NaCl).

         Thank you for your comments. As you suggested, as additional experiments, we measured the particle size of NI-LNs dispersed   in PBS. The mean particle size was 59.4 nm and almost same as the result of   NI-LN in distilled water. However, as you pointed out, we agree with your   idea that dispersion medium affects the physicochemical properties of NI-LNs.   Actually, as new experiments, we have been evaluating the physicochemical   properties of LNs in simulated gastrointestinal fluids (Fasted State Simulated Intestinal Fluid (FaSSIF) or Fed State Simulated Intestinal Fluid (FeSSIF)),   and we want to report them in the near future as a further study. We believe   you understand our thought.

6

Tables 5 and 6 – the type of ratio   (molar) should be reported.

        Thank you. The word “HSPC/DPPG (molar ratio)” has been added in revised Table 5 and 6 in the revised   manuscript.  

7

The quality of Figure 8 panel B must   be improved.

        We agree with your idea. However, it would be difficult to additionally take cryo-TEM images and replace new one for the following reasons. One is revision due of this manuscript is within only one week. Because this cryo-TEM machine is not mine and common use apparatus, we cannot use this   machine in this period. In addition, as another reason, even if we can use   this machine and perform additional experiments, as you know, it would spend a   huge amount of time for taking the images by cryo-TEM, because it is technically   difficult to observe such kind of small enough particle. In this image (Fig. 8B), to easily find and focus on these small particles, the magnification was a little bit decreased and to   justify the length of scale bar of other images, the image was expanded, which might become blurred. However, we think this image was still enough to   show the morphology of these small particles. So, we hope reviewer understand our situation and thought.

Reviewer 2 Report

        In the manuscript “Combination of roll grinding and high-pressure homogenization can prepare stable bicelles for drug delivery” the authors reported the synthesis and characterization of bicelles for nifedipine and phenytoin delivery.

Although the paper is well written, I think that the authors confused the several concepts, such as nanoparticles, bicelles and liposomes. In my opinion, the manuscript is very unclear. 

        Nanoparticles are solid materials in nanoscale and bicelles are amphipathic structures composed of two phospholipids having a critical micelle concentration to be formed. The presence of the phospholipid confers high fluidity of the lipids. Thus, the two concepts (nanoparticles and bicelles) are not the same thing. I suggest that the authors replace “lipid nanoparticles” for “lipid nanostructures” or “lipid aggregates”

        Another negative point is that the authors indicate that the authors said in Abstract that “The NI-LNs exhibited high entrapment efficiency, long-term stability, and enhanced NI bioavailability”. In the present paper, the authors did not present any in vivo studies for bioavailability. Therefore, I suggest that the author remove this affirmation from the manuscript. 

        My main commentaries are given below:

        Abstract

        The abstract is very confusing. In the beginning of Abstract, the author said “To improve the solubility of the drug nifedipine (NI), NI-encapsulated lipid nanoparticles (NI-LNs) were prepared from neutral hydrogenated” However, in the final of the Abstract, the authors said about other dug: phenytoin. Please, reformulate it. 

        The authors said that the NI-LNs enhanced the bioavailability. I think this is not true. The authors did not show any data for bioavailability studies. Please, remove it.

        Introduction

        (Line 69-73) I think that the reference (27) is not so recent. Please, rephrase the sentence.

        Material and Methods 

        (Line 126) Please, supply the real temperature, the angle and the concentration of the formulations that were measured in DLS. How many measurements of each sample did the authors do?

Did the authors dilute the samples before the analysis?

       Results

       (Line 188) Liposomes are not particles. Liposomes are vesicles in nanoscale. Please, rephrase the sentence. 

       (line 206) How can the authors affirm that the mean particle size became slightly bigger after freeze-drying, but the bicelle structure was preserved”?  Please, clarify it. 

        In addition, the authors said that “Thus, freeze-drying and the addition of a lyophilization stabilizer did not affect the NI-LN structure”. How the lyophilization stabilizers did not affect the NI-LN structure if the particle size became bigger? For me it is not clear. 

       How did the author measure the size relating to particles, micelles and bicelles as indicated inTable 1, 2 and 3?

       (Line 271) Liposomes? Please, explain it for better understanding. 

       Author Response

  We thank the reviewers for the very valuable comments. We have addressed all the issues raised by the reviewers in a point-by-point manner as noted below. The changes made in the manuscript have been marked in red.

      Reviewer #2. We thank the reviewer for his or her useful comments.

1. Although   the paper is well written, I think that the authors confused the several   concepts, such as nanoparticles, bicelles and liposomes. In my opinion, the   manuscript is very unclear.

Nanoparticles   are solid materials in nanoscale and bicelles are amphipathic structures   composed of two phospholipids having a critical micelle concentration to be   formed. The presence of the phospholipid confers high fluidity of the lipids.   Thus, the two concepts (nanoparticles and bicelles) are not the same thing. I   suggest that the authors replace “lipid nanoparticles” for “lipid   nanostructures” or “lipid aggregates”

Thank   you for your comment. We agree with your idea. As you pointed out, definition   of the lipid-based nanocarriers must be very important. Lipid-based nanocarriers   are broadly comprised of liposomes, niosomes, cubosomes, nanoemulsions,   nanomicelles, and lipid nanoparticles (LNs) (Gan et al., Drug Discov. Today. 2013). We think that “solid lipid nanoparticles (SLN)” is comprised of LNs and the   SLN structure is composed of a solid lipid core, which may contain   triglycerides, glyceride mixtures, or waxes that are solid at both room   temperature and human body temperature (Wissing SA, et al., Solid lipid   nanoparticles for parenteral drug delivery. Adv Drug Deliv Rev. 2004). From   this perspective, although two phospholipids used in this study are solid at   both room temperatures, we doubt that whether our nanocarrier is SLN.   Therefore, we used the term “Lipid nanoparticles (LNs)” which means “lipid-based   nanoparticles” until now (Ohshima, H. et al., Int. J. Pharm. 2009, Funakoshi,   Y. et al., Int. J. Pharm. 2013,). In addition, throughout a   series of experiment in this study, we found for the first time, that these   lipid-based nanoparticles were defined as bicelles. Therefore, we think the   term “lipid aggregates” as you suggested is also good, but we want to use the   word “LNs” after revising the meaning from “lipid nanoparticle” to “lipid-based   nanoparticles” in the revised manuscript as follows.

(Abstract)

: To   improve the solubility of the drug nifedipine (NI), NI-encapsulated lipid-based nanoparticles (NI-LNs) have been prepared

We   hope reviewer understand our thought. If our thought is something wrong,   please let me know again. Thank you very much for your useful comments.

2. The   abstract is very confusing. In the beginning of Abstract, the author said “To   improve the solubility of the drug nifedipine (NI), NI-encapsulated lipid   nanoparticles (NI-LNs) were prepared from neutral hydrogenated” However, in   the final of the Abstract, the authors said about other dug: phenytoin.   Please, reformulate it.

The authors   said that the NI-LNs enhanced the bioavailability. I think this is not true.   The authors did not show any data for bioavailability studies. Please, remove   it.

We are so sorry   for confusing you. In the previous studies, lipid nanoparticles have already   been prepared to enhance the solubility of poorly water-soluble drug. These   nanoparticles have exhibited high entrapment efficiency, long-term stability,   and enhanced NI bioavailability; however, it remains unclear about the   structure.  In this study, to better   understand their structures, cryo transmission electron microscopy and atomic   force microscopy were performed. So, at the beginning part of our abstract,   the descriptions must be confusing. Therefore, these descriptions have been   revised as follows. Thank you so much.

(Abstract)

: To improve   the solubility of the drug nifedipine (NI), NI-encapsulated lipid-based nanoparticles   (NI-LNs) have been prepared from neutral   hydrogenated soybean phosphatidylcholine and negatively charged dipalmitoylphosphadylglycerol   at a molar ratio of 5/1 using by roll grinding and a high-pressure   homogenization. The NI-LNs have exhibited high   entrapment efficiency, long-term stability, and enhanced NI bioavailability.   To better understand their structures, cryo transmission electron microscopy   and atomic force microscopy were performed in the   present study.

3. (Line 69-73) I think that the   reference (27) is not so recent. Please, rephrase the sentence.

That’s true. Based   on the reviewer’s comment, the following description has been revised in the   revised manuscript.

(Page 2, line 70)

: Previously, we reported that the water-solubility of....

Comment #4

        (Line 126) Please, supply the real temperature, the angle and the concentration of   the formulations that were measured in DLS. How many measurements of each sample did the authors do?

Did the authors dilute the samples before the analysis?

         Response: 

       Mean particle size, PDI and zeta potential were   measured   by zetasizer at a scattering angle of 90° at room temperature   (25°C). In addition, the values of mean count rate of intact samples   were 200–500 kcps   which are suitable for this measurement; therefore, we did not perform the dilution in this study. And we repeated   measurement three times per a sample.  

       Based on the reviewer’s comment, the following sentence has been revised in the revised manuscript.
      (Page 3, line 130-)

         :The mean particle size and the zeta potential of LN suspensions were measured with dynamic light scattering (DLS) (Zetasizer nano ZS, Malvern Instruments Ltd., Worcestershire, UK) at a scattering angle of 90° at room temperature (25°C). Mean particle sizes were based on the scattering intensity, while the zeta potential was based on electrophoretic mobility. Since the values of mean count rate of the samples were 200–500 kcps which are suitable for this measurement, intact samples without dilution were used in this measurement. We repeated measurement three times per a sample. 

5. (Line   188) Liposomes are not particles. Liposomes are vesicles in nanoscale.   Please, rephrase the sentence.

Thank  you for your comment. Although we think that liposomes are one of lipid-based  nanoparticles in a broad sense, as you pointed out, liposomes are definitely vesicles  in nanoscale.

Based on your comments, the following description has been revised in the revised   manuscript.

 (Page 5, line 195)

:  Spherical vesicles such as liposomes sometimes strongly

6. (line   206) How can the authors affirm that the mean particle size became slightly   bigger after freeze-drying, but the bicelle structure was preserved”?  Please, clarify it.

In   addition, the authors said that “Thus, freeze-drying and the addition of a   lyophilization stabilizer did not affect the NI-LN structure”. How the   lyophilization stabilizers did not affect the NI-LN structure if the particle   size became bigger? For me it is not clear.

How   did the author measure the size relating to particles, micelles and bicelles   as indicated inTable 1, 2 and 3?

       Thank you for your comment and sorry for insufficient explanation. Firstly, as for   the description (line 206), we judged this by just watching the Fig. 4B compared with Fig. 1 and 4A (just visual observation). But, after this description, we   have already checked the mean particle size determined by cryo-TEM images (Table 1) and zetasizer using DLS (Table 2), and in fact, mean particle   sizes were a little bit bigger than others.

  Based on the reviewer’s comments, this description seems to be confusable; so, the   description has been revised as follows.

(Page   6, line 214) : The particle size looked slightly bigger after freeze-drying,   but the bicelle structure was preserved.

An   increase in particle size might be explained by insertion of sucrose  molecules in the phospholipid molecules of the lipid bilayer. However, as   shown in Fig. 4B, the disk   structure of the bicelles was maintained after reconstitution. Namely, we   cannot observe any structural changes from disk-like to spherical or other   forms, for example. Therefore, we thought that   freeze-drying and the addition of a lyophilization stabilizer did not affect   the NI-LN structure itself. Actually, we have no idea why the particle size became slight bigger; but we speculate that some aggregation between   comparatively big and small particles would occur. We hope that reviewer  understand our thought.

In  addition, one hundred particles in cryo-TEM images printed on paper were   measured with a ruler to determine the size relating to particles. So, the   following description has been revised in the revised manuscript. Thank you  for your comments.

       (Page 6, line 219)

      : one hundred particles in cryo-TEM images printed   on paper were measured with a ruler (Table 1).

7. (Line 271) Liposomes? Please, explain it for better   understanding.

       As shown in Fig. 6A and 6B, the shape   did not change when the sample stage was rotated by 20 degrees; so the   particles were thought not to be bicelles or flat particles. When looking at inner   part of the particles, the particles were seems like to be liposomes having internal   water phase because inner part looked transparent. If the particles were not   liposomes, the inner part of the particles would look black. Furthermore, as   we have already given the explanation (line   354-3555 in the original manuscript), it was reported that cholesterol-free   liposomes became polygonal particle (#46 in the original manuscript).   Therefore, we think this particle in nanoscale might be liposomes.

Based on the reviewer’s comment, the same reference has been   added in the revised manuscript.

        (Page 8, line 280)

       :Thus, the particles were not flat and could have been polygonal liposomes [47].

Reviewer 3 Report

Dear Editor,

        The paper by Matsuoa et al. entitled “Combination of roll grinding and high-pressure homogenization can prepare stable bicelles for drug delivery” comprehensively elucidates how these carriers improve the delivery some drugs. The subject fits to the journal scope and, in my opinion, is interesting for the scientific community.

        This work provides an useful method for stable bicelles for drug delivery.

        It needs a small revision of some parts. The enhanced solubility of water insoluble drugs seem to be interesting. Introduction part is well written and provided by proper references. The paper substantially deals with the characterization of these nanoparticles.

From the scientific point of view I have just one objection. Afterwards, only a formal issue.

        Major request:

        DLS measurements

        The authors along the manuscript speak about the mean particle sizes based on the scattering intensity.

        Please be a little more precise and clearer for researchers not dealing with DLS measurements. Specify that you are reading the Z-average, if I am true, and that this measure is speaking about the hydrodynamic radius or diameter of the particles and micelles. Please specify that usually this hydrodimanic radius is a little bigger than the particles itself by quoting the proper papers……. and so on.

For instance the sentences written from line 216 to line 218 and in line 288 are for sure correct but seems to me too simplistic.

        Minor request:

        LINE 177: “These changes indicated flat particles. “ (Please quote proper reference)

        Based on the above arguments, I recommend the paper for publication.

       Author Response

  We thank the reviewers for the very valuable comments. We have addressed all the issues raised by the reviewers in a point-by-point manner as noted below. The changes made in the manuscript have been marked in red.

        Reviewer #3. We thank the reviewer for his or her positive and useful comments.

Comments

Our   response

1

DLS   measurements

The   authors along the manuscript speak about the mean particle sizes based on the   scattering intensity.

Please be   a little more precise and clearer for researchers not dealing with DLS   measurements. Specify that you are reading the Z-average, if I am true, and   that this measure is speaking about the hydrodynamic radius or diameter of   the particles and micelles. Please specify that usually this hydrodimanic   radius is a little bigger than the particles itself by quoting the proper   papers……. and so on.

For   instance…. the sentences written from line 216 to line 218 and in line 288   are for sure correct but seems to me too simplistic.

        Thank you so much for your comments. As you pointed out, it is known that mean particle   size measured by scattering intensity from brownian motion is bigger than   that of measured by other methods. Because bigger particles can influence scattering   intensity, DLS has reported to tend to overestimate the hydrodynamic diameter   than real one (Magdalena, W. et al., Colloids and Surfaces B: Biointerfaces   2014). Moreover, DLS furnishes the hydrodynamic diameter depending on   assumption that the particle is sphere (Doroty, C. et al., Int. J. Pharm. 2015), but the bicelles in this study were not   spherical; so we thought the possibility that the data from DLS might be not accurate. Thus, we think measurement by only DLS seems not to   be sufficient and measurement by cryo-TEM would also be necessary.

Based on the   reviewer’s comments, the following descriptions have been revised and new references   have been added to the revised manuscript.  

        (Page 6, line   218)

        :Cryo-TEM and DLS   data were based on the number reference and on scattering intensity from   brownian motion, respectively. Previously, it was reported that DLS tends to   overestimate the hydrodynamic diameter than real one because large particles   strongly influences scattering intensity [44]. In addition, DLS furnished the   hydrodynamic diameter depending on assumption that the particles are sphere   [45], but the bicelles in this study were not spherical. Therefore, the   difference of measurement principle would be involved in the differences of   particle size.

        (References)

44.  Magdalena, W.; Jan, B.; Maria, N.; Paweł,   W.; Mariusz, K. Interactions of serum with polyelectrolyte-stabilized   liposomes: Cryo-TEM studies. Colloids   and Surfaces B: Biointerfaces 2014,   120, 152-159.

45.  Doroty, C.; Jonathan, C.; Jenna, B.; William,   J. M.; Maria, A. O.; Sheng, Q. Disc-shaped polyoxyethylene glycol glycerides   gel nanoparticles as novel protein delivery vehicles. Int. J. Pharm. 2015, 496, 1015-1025.

2

LINE 177: “These changes indicated flat particles. “ (Please   quote proper reference)

Thank you for   your comment. The following description has been revised and new reference has   been added in the revised manuscript.

        (Page 4, line 184)

        :These changes indicated flat particles as previously reported [39]

        (Reference)

39.  Yasuhara, K.; Miki, S.; Nakazono, H.; Ohta,   A.; Kikuchi, J. Synthesis of organic–inorganic hybrid bicelles–lipid bilayer   nanodiscs encompassed by siloxane surfaces.Chem. Commun., 2011, 47, 4691–4693.

Round  2

Reviewer 1 Report

        The authors properly discussed the proposed crticisms in the revised version.

        The manuscript is worthy of publication.

Reviewer 3 Report

Dear editor,

        the authors have done a great job answering correctly all my criticisms and by adding proper references.

        Please accept in this form.